# Lattice-Based Certificateless Proxy Re-Signature for IoT: A Computation-and-Storage Optimized Post-Quantum Scheme

**DOI:** 10.3390/s25154848

**Published:** 2025-08-06

**Authors:** Zhanzhen Wei, Gongjian Lan, Hong Zhao, Zhaobin Li, Zheng Ju

**Affiliations:** Department of Electronic and Communication Engineering, Beijing Electronic Science and Technology Institute, Beijing 100070, China; wzz@besti.edu.cn (Z.W.); zh@besti.edu.cn (H.Z.); lzb@besti.edu.cn (Z.L.); 20233801@mail.besti.edu.cn (Z.J.)

**Keywords:** lattice, certificateless, proxy re-signature, Dilithium, post-quantum, Internet of Things (IoT), authentication

## Abstract

**Highlights:**

**What are the main findings?**
We propose a novel and practical proxy re-signature scheme that utilizes the Dilithium algorithm.Performance evaluation and comparative analysis demonstrate that our proposed scheme significantly reduces the computational overhead compared to existing algebraic lattice-based schemes. Furthermore, it substantially optimizes the signature storage size compared to mainstream NTRU-based solutions, achieving a 52.9% reduction in storage space.

**What are the implications of the main findings?**
This study fills a critical research gap by establishing a practical certificateless proxy re-signature scheme based on the Dilithium algorithm. Beyond this novelty, it also promotes the broader application and exploration of the Dilithium algorithm within the domain of certificateless cryptography.The considerable reduction in computational overhead positions our proposed scheme as a highly attractive option for resource-constrained environments, such as Internet of Things (IoT) terminals.

**Abstract:**

Proxy re-signature enables transitive authentication of digital identities across different domains and has significant application value in areas such as digital rights management, cross-domain certificate validation, and distributed system access control. However, most existing proxy re-signature schemes, which are predominantly based on traditional public-key cryptosystems, face security vulnerabilities and certificate management bottlenecks. While identity-based schemes alleviate some issues, they introduce key escrow concerns. Certificateless schemes effectively resolve both certificate management and key escrow problems but remain vulnerable to quantum computing threats. To address these limitations, this paper constructs an efficient post-quantum certificateless proxy re-signature scheme based on algebraic lattices. Building upon algebraic lattice theory and leveraging the Dilithium algorithm, our scheme innovatively employs a lattice basis reduction-assisted parameter selection strategy to mitigate the potential algebraic attack vectors inherent in the NTRU lattice structure. This ensures the security and integrity of multi-party communication in quantum-threat environments. Furthermore, the scheme significantly reduces computational overhead and optimizes signature storage complexity through structured compression techniques, facilitating deployment on resource-constrained devices like Internet of Things (IoT) terminals. We formally prove the unforgeability of the scheme under the adaptive chosen-message attack model, with its security reducible to the hardness of the corresponding underlying lattice problems.

## 1. Introduction

Proxy re-signature schemes provide a mechanism for the transitive authentication of cross-domain digital identities. In proxy re-signature, a proxy (P) serves the function of transforming signatures on an identical message between two distinct signers. Specifically, using a proxy re-signature key, proxy P can convert delegator A’s signature on a given message into a signature verifiable with delegatee B’s public key for that same message. This transformation ensures that the resulting re-signed signature is indistinguishable from one genuinely generated by delegatee B while being clearly distinguishable from delegator A’s original signature. Proxy re-signature technology has significant applications in diverse fields, such as Digital Rights Management (DRM), cross-domain certificate interoperability, and access control in distributed systems. A prime example of this is its application in international travel document systems. Consider a traveler (C) holding an electronic signature (${Sig}_{E}$) issued by their home country (E) who seeks entry into country F. The border control agency in country F, acting as a proxy entity, first verifies the validity of ${Sig}_{E}$. Upon successful verification, the agency can convert ${Sig}_{E}$ into an electronic signature (${Sig}_{F}$) that conforms to country F’s standards. Subsequently, relevant authorities within country F can perform the traveler’s identity verification and achieve transnational authentication merely by utilizing the public key of country F’s border control agency. This process obviates the need to manage and maintain complex, transnational certificate chains.

The concept of proxy re-signature was first introduced by Blaze et al. [1]. Ateniese and Hohenberger [2] formally defined the security model and explored additional application scenarios. Early applications primarily focused on smart card key updates, enabling the dynamic expansion of terminal device key spaces using proxy signatures. Subsequently, their use has expanded to areas such as anonymous group signatures and distributed system path attestation. For instance, in distributed network routing verification, data packets can leverage a proxy signature chain to prove their complete transmission path, with verifiers only requiring the public key of the terminal node. However, these early proxy re-signature schemes were predominantly based on certificate-based public-key cryptosystems. In such systems, the public keys of the delegatee and delegator must be obtained from certificates prior to signature verification, leading to significant certificate distribution and management challenges. To address the bottleneck of certificate management, researchers [3,4] have developed various identity-based proxy re-signature (IBPRS) schemes. These schemes utilize user identity information as public keys, thereby avoiding the reliance on certificates.

Nevertheless, in IBPRS schemes, the private keys of both the delegatee and delegator are generated by a Private Key Generator (PKG). This inherently introduces a key escrow problem [5], as the PKG possesses knowledge of all user private keys, enabling it to potentially eavesdrop on communications or to forge user signatures. To resolve both certificate management and key escrow issues, scholars have begun investigating certificateless proxy re-signature (CLPRS) schemes. Guo et al. [6] combined certificateless cryptography with proxy re-signature, proposing the first bidirectional CLPRS scheme. Although this approach eliminated certificate dependency and avoided the key escrow problem, a concrete security proof was not provided. The proxy re-signature schemes discussed previously are all based on traditional public key cryptography, with their security relying on the presumed intractability of problems such as large integer factorization and discrete logarithm problems. This reliance introduces inherent security vulnerabilities to the system. With the advent of quantum computing, the security of public-key cryptosystems founded on classically hard number-theoretic problems faces a significant challenge. As Shor [7] demonstrated, the advancement of quantum computing renders problems like discrete logarithm and integer factorization computationally tractable. Thus, developing post-quantum certificateless proxy re-signature schemes is the only comprehensive approach to resolving the certificate management bottleneck, key escrow problem, and imminent quantum computational threat faced by current proxy re-signature technologies.

To address these problems, we introduce a computationally efficient post-quantum certificateless proxy re-signature scheme based on algebraic lattices. This scheme is designed to guarantee the security and integrity of communication messages exchanged among the Key Generation Center (KGC), proxy, delegator, and delegatee, even in quantum attack environments.

## 2. Contribution

The main contributions of this paper are as follows:(1)We propose an efficient certificateless proxy re-signature scheme with a core architecture based on algebraic lattice theory. This scheme leverages the Dilithium algorithm as a foundational building block, ensuring operational efficiency while resolving the complexities associated with Discrete Gaussian Sampling. In contrast to comparable schemes, our proposal innovatively adopts a parameter-selection strategy enhanced by lattice basis reduction. This strategy ensures strict adherence to the NIST-standardized parameter configurations while effectively circumventing the potential algebraic attack vectors inherent in NTRU lattice structures.(2)By applying structured compression techniques, our scheme optimizes the storage size of signatures to be comparable to that of NTRU-based architectures. This results in a storage space compression rate of over 52.9% when compared to mainstream NTRU-based signature schemes, offering marked advantages for deployment in storage-constrained environments, including Internet of Things (IoT) terminals.(3)We formally prove our scheme to be unforgeable under the Existential Unforgeability under Chosen Message Attack (EUF-CMA) security model, based on the dual hardness assumptions of the Module Small Integer Solution (MSIS) and Module Learning With Errors (MLWE) problems. The security of the scheme is reducible to the presumed intractability of these underlying mathematical problems.

## 3. Related Work

Current research hotspots in proxy re-signature primarily encompass identity-based schemes, lightweight schemes for the Internet of Things (IoT), certificateless schemes, and post-quantum secure schemes. Dutta et al. [8] proposed an identity-based unidirectional proxy re-signature scheme; however, it is susceptible to private key leakage. Tian et al. [9] constructed an identity-based bidirectional proxy re-signature scheme. However, its proxy re-signature process necessitates joint computation involving the private keys of both the delegator and delegatee, thereby increasing the key management complexity. Zhang et al. [10] introduced an identity-based non-interactive proxy re-signature scheme tailored for Mobile Edge Computing. While this scheme reduces the computational overhead by avoiding bilinear pairing operations, its reliance on the hardness of large integer factorization renders it vulnerable to quantum computing threats. In the context of mobile Internet environments, Lei et al. [11] proposed a unidirectional, variable-threshold proxy re-signature scheme notable for its shorter signature lengths, reduced computational costs, enhanced verification efficiency, and improved adaptability; this construction was also formally proven secure in the Standard Model against both collusion and adaptive chosen message attacks. Nevertheless, its reliance on the hardness of the bilinear pairing problem renders it incapable of withstanding quantum computing attacks.

Operating within certificateless frameworks, Fan et al. [12] remedied the limitations inherent in the signature protocol devised by Tian et al. [9]. This resolution involved the presentation of a certificateless proxy re-signature method exhibiting superior operational efficiency and distinguished by more concise private keys. In a separate study, Wu et al. [13] proposed a flexible unidirectional certificateless proxy re-signature scheme. Nevertheless, the structural dependence of this particular scheme on the bilinear pairing problem means that it is not equipped to counteract threats emerging from quantum computation. Zhang et al. [14] developed a revocable certificateless proxy re-signature scheme capable of supporting signature evolution within Electronic Health Record (EHR) sharing systems. Specifically, it facilitates dynamic user management and enables efficient revocation and updating of signatures in response to evolving data requirements. However, this scheme also lacks resistance to quantum computing threats.

Currently, post-quantum signature methodologies predominantly fall into three main classes: those derived from hash functions, those constructed using multivariate polynomials, and those based on lattices. The security of hash-derived signatures is contingent upon the collision resistance of the underlying hash functions; however, such schemes often present limitations regarding both signature compactness and execution velocity. Multivariate polynomial signatures, on the other hand, are recognized as being vulnerable to algebraic cryptanalysis, which can potentially undermine their claimed security. Diverging from these two paradigms, lattice-based signatures exhibit notable strengths in terms of computational performance and security robustness. The initial basis for public-key cryptography leveraging lattices was provided by Ajtai [15], who demonstrated a fundamental linkage between the average-case difficulty and worst-case intractability of certain lattice problems. Following this foundational work, Gentry [16], during the process of developing signature schemes from lattices, introduced the precise notion of a ‘one-way trapdoor function.’ In [17], Lyubashevsky introduced a novel methodology for converting identification schemes into signature schemes using the Fiat-Shamir transform [18,19]. This approach incorporates an ‘abort’ mechanism to discard any signature values that might leak private key information, thereby ensuring that the output signature values adhere to a uniform distribution. Later, Lyubashevsky [20] employed rejection sampling techniques to generalize sampling to arbitrary distributions and demonstrated that signature schemes based on the Learning With Errors (LWE) problem could achieve smaller public key sizes.

Building upon Lyubashevsky’s signature scheme [20], Tian et al. [21] introduced a lattice-based certificateless signature scheme that is notable for its shorter private keys and enhanced efficiency compared with other contemporary schemes. Subsequently, Xie et al. [22] proposed a versatile unidirectional lattice-based proxy re-signature scheme. However, a significant drawback of their approach is that users must fully generate their own public and private keys, creating vulnerabilities to attacks from malicious users. Furthermore, their scheme risks exposing the delegatee’s private key during the generation of the re-signature key. Fan et al. [23] developed a lattice-based re-signature method proven secure in the CCA-PRE model. Nonetheless, this scheme is susceptible to man-in-the-middle attacks and requires a greater storage capacity for re-signatures. Jiang et al. [24] put forward a lattice-based proxy re-signature scheme that permits a message to be re-signed multiple times. A proxy re-signature construction employing lattice structures, designed for unidirectional applications and infinite use, was proposed by Chen et al. [25]. This scheme incorporates private re-signature keys, which allow an individual message to undergo an unbounded number of re-signing operations. Separately, Luo et al. [26] advanced a proposal for an attribute-based proxy re-signature methodology, establishing its foundations upon conventional lattice structures. Through the application of dual-mode cryptographic techniques, Zhou et al. [27] developed a certificateless proxy re-signature scheme engineered to be resistant to collusion attacks; however, an efficiency analysis of this scheme was not provided, and it suffers from large signature sizes.

To date, scholarly investigations into certificateless proxy re-signature schemes based on algebraic lattices are not extensive. This scarcity is primarily because certificateless signature schemes constructed using algebraic lattices often result in large signature or private key lengths, thereby imposing a notable burden on the available storage resources. To address the challenge of substantial signature lengths, Guneysu et al. [28] proposed a methodology centered on partitioning numerical values into two distinct components: higher-significance bits and lower-significance bits. This approach permits the elision of the lower-significance bits on the condition that their removal does not alter the rounding outcome of the higher-significance bits, consequently leading to diminished storage requirements. Concurrently, Bai et al. [29] introduced a method for discarding signature subcomponents, which implicitly incorporates a proof of noise knowledge within the overarching proof pertaining to the private key. Recognized as a NIST-standardized algorithm for post-quantum signatures, Dilithium utilizes compression strategies comparable to those in Guneysu’s work. It further makes use of ‘hints’ within its rounding operations, thereby aiming to avert failures during the verification process. Furthermore, Dilithium is implemented on algebraic lattices, and its security is based on the presumed hardness of the Module Small Integer Solution (MSIS) and Module Learning With Errors (MLWE) problems [30]. Due to its design, which incorporates uniform key sampling, the scheme demonstrates resistance to known algebraic and subfield attacks. For NIST-selected parameter sets of Dilithium (e.g., security levels 2, 3, and 5), solving the corresponding MLWE and MSIS instances remains computationally infeasible under currently known optimal classical and quantum attacks [31]. In summary, existing certificateless proxy re-signature schemes are typically either designed based on traditional computationally hard problems, rendering them vulnerable to quantum attacks, or they achieve quantum security at the cost of low storage and computational efficiency, making them unsuitable for practical deployment. Consequently, there is substantial value in furthering the design of certificateless proxy re-signature schemes employing lattices that concurrently achieve high efficiency and strong security guarantees.

## 4. Preliminaries

### 4.1. Parameter Notation and Their Definitions

Table 1 summarizes the notation used in the proposed scheme.

### 4.2. Lattices

Given an n-dimensional space, let $b_{1},\ldots ,b_{n}$ be m linearly independent vectors. Suppose matrix $B\in R^{n*m}$ is formed by these vectors. Thereafter, let $L\left(B\right)=\{Bx:\;x\in Z^{m}\}$ denote the lattice that B generates.

### 4.3. Hardness Assumption

The conceptual integration of module lattices facilitates the formulation of the Module Learning With Errors (MLWE) and Module Small Integer Solution (MSIS) problems, which represent structural advancements over their foundational counterparts, Learning With Errors (LWE) and Small Integer Solution (SIS), respectively. These intractability assumptions, functioning as broader versions of LWE and SIS, are fundamentally derived from challenges in standard lattice theory. The primary application of the MLWE assumption lies in safeguarding cryptographic keys and offering resilience against attacks aimed at key recovery. In tandem, the postulated hardness of the MSIS problem forms the cryptographic basis for the robust unforgeability of the signature scheme. A comprehensive exposition of these foundational hardness assumptions is allocated to the security analysis chapter later in this document.


**Definition 1.** 

*SIS problem:*
*Given a positive integer q, and a uniformly random matrix $A\in Z_{q}^{n\times m}$**, find a non-zero vector $v\in Z^{m}$* *such that Av=0 mod q* *and ‖v‖≤β*.



**Definition 2.** 

*MSIS problem:*
*For a given matrix *$A\in Z_{q}^{m\times k}$*, the MSIS challenge is to find a non-zero integer vector v satisfying the conditions * [A|I]⋅v=0(modq) *and* ‖v‖≤β.



**Definition 3.** 

*MLWE problem:*
*A matrix* $A\leftarrow R_{q}^{n\times m}$ *is selected uniformly at random. Subsequently, two vectors,* $s_{1}$ *and* $s_{2}$*, are sampled according to* $s_{1}\leftarrow \chi^{m}$ *and* $s_{2}\leftarrow \chi^{n}$*, where* χ *represents a probability distribution over a ring R. The output is* $t\;=\;As_{1}+s_{2}\in R_{q}^{n}$*. The decision version of MLWE is to distinguish between the pairs* $\left(A,t\;=\;As_{1}+s_{2}\right)$ *and* $\left(A,\;t\right)\leftarrow R_{q}^{n\times m}\times R_{q}^{n}$.


### 4.4. Number Theoretic Transform (NTT) Domain Representation

Let $a\in R_{q}$ (denoted as $\left(â\in Z_{q}^{256}\right)$) be a polynomial. Its representation in the NTT domain, denoted by â, is specified as the vector: $â=(a\left(r_{0}\right),\;a\left(-r_{0}\right),\ldots ,\;a\left(r_{127}\right),a\left(-r_{127}\right))$. In this formulation, the components $r_{i}$ are defined by the expression $r_{i}=r^{brv(128+i)}$. The function brv(k) herein signifies the bit-reversal of the 8-bit integer k.

## 5. Construction of the Scheme

### 5.1. System Model

Our proposed scheme design targets general proxy re-signature requirements. To intuitively demonstrate its application value, this section constructs a system model using the Medical Internet of Things as an example scenario. The system architecture, illustrated in Figure 1, comprises four distinct entities: the Key Generation Center (KGC), Terminal User (TU), Cloud Server (CS), and Healthcare Authority (HA).

(1)Key Generation Center (KGC): A trusted root authority entrusted with initializing the system, managing the registration of TUs and HAs, and distributing partial private keys.(2)Terminal User (TU): An entity representing the data owner and their associated medical devices, characterized by limited computational and storage resources. The TU obtains system parameters and a partial private key from the KGC upon registration, after which it digitally signs personal medical data (e.g., EHRs) for transmission to the CS.(3)Cloud Server (CS): A cloud service platform designed for the storage, processing, and dissemination of the TU’s health data. The CS validates the cryptographic signature of incoming data from the TU before ingestion. Upon a valid request from an HA, the CS executes a proxy re-signature protocol to delegate access.(4)Healthcare Authority (HA): A service provider that must also be registered with the KGC is responsible for performing authentication and validation checks. These checks confirm the data’s authenticity (originating from the TU), integrity (unaltered content), and access legitimacy (explicit authorization from the TU).

Within this framework, the TU generates personal health records from wearable physiological monitors and delegates access to an HA for diagnostic use. The operational protocol involves the following phases:

Phase 1: System Initialization and Entity Registration. The KGC generates a master key pair and public system parameters. Entities petition the KGC for registration, and the KGC generates a unique partial private key for each entity and delivers it via a secure channel.

Phase 2: Full Key Derivation and Source Signature. Each entity derives its full key pair by combining its self-generated secret value with the partial key obtained from the KGC. The TU then applies an original signature to its data packet and securely uploads it to the CS for storage. The CS can verify the validity of the original signature, thereby ensuring integrity during transit, but lacks the ability to decipher the signature’s content or forge new signatures. Figure 2 illustrates the concrete workflow.

Phase 3: Authorization and Re-Signature Key Generation. When the TU decides to grant access to an HA, it executes a re-signature key generation algorithm to produce a key that is designated for that HA. This key is securely transmitted to the CS, which stores it in association with the TU data.

Phase 4: Proxy Re-Signature and Data Access. The HA initiates a data access request by invoking a re-signature key generation algorithm to produce a key designated for the TU and sends it to the CS. The CS verifies the identity of the HA, retrieves the TU’s data packet along with the key designated for that HA, executes the re-signature algorithm to convert the original signature into a re-signature, and transmits the resultant data packet to the HA. This phase ensures delegation unforgeability (effective re-signature keys cannot be forged without the original signer’s authorization) and delegation precision (re-signature keys are not misused by others).

Phase 5: Data Verification and Utilization. The HA validates the re-signature. A positive verification result cryptographically assures the following properties:(1)Authenticity: The data originated and was signed by the claimed TU.(2)Integrity: The data have remained unchanged since their creation.(3)Delegation unforgeability: Access by the HA is explicitly granted by the TU.(4)Long-Term Quantum Resistance: The security of historical signatures is preserved against retrospective attacks launched by future quantum computers.

### 5.2. Basic Functions

#### 5.2.1. Hash

Our proposed scheme utilizes several distinct algorithms to map strings to elements within diverse domains. The algorithms are detailed below.

Collision-Resistant Hash (CRH): Maps strings to $\;{\left\{0,\;1\right\}}^{384}$.

ExpandMask: Maps a string of any length to a vector $y\in S_{\gamma 1}^{l}$.

ExpandA: Maps a uniform random seed $\rho \in {\left\{0,\;1\right\}}^{256}$ to a matrix $A\;\in \;R_{q}^{k\times l}$, represented in the NTT domain with coefficients in $Z_{q}^{256}$.

ExpandS: Maps a uniform random seed $\rho^{\prime }\in {\left\{0,\;1\right\}}^{256}$ to vectors $s_{1}\leftarrow S_{\eta }^{l}$ and $s_{2}\leftarrow S_{\eta }^{k}$, each with coefficients in the interval [−η,η].

Moreover, we use the function and its parameters as defined in reference [31]: ExpandA make use of the more efficient extendable-Output Function (XOF) $H_{128}$, whereas ExpandS and ExpandMask use the XOF H.

#### 5.2.2. SampleInBall

Generates an element $B_{\tau }$ of pseudo-randomly using the seed ρ. The first 8 bytes of H(ρ) are used to choose the signs of the nonzero entries of c, and subsequent bytes of H(ρ) are used to choose the positions of those nonzero entries. The parameter τ is always less than or equal to 64, and thus 8 bytes are suffcient to choose the signs for all τ nonzero entries of c.

#### 5.2.3. Element Decomposition

The basic idea is to drop the d low-order bits of each coefficient of the polynomial vector t from the public key using the function Power2Round. The procedure detailed in Algorithm 1 facilitates the partitioning of an element $r\;\in \;Z_{q}$ into its higher-order bits $r_{1}$ and lower-order bits $r_{0}$. The element r can be represented as $r=\;r_{1}\cdot 2^{d}\;{+\;r}_{0}$, where $r_{0}=r\;mod\;2^{d}$ and $r_{1}=(r\;-\;r_{0})∕2d$. Here, d represents the number of binary bits. To ensure that the deviation in the value represented by the higher-order bits does not exceed a magnitude of one.
**Algorithm 1** 
${Power2Round}_{q}(r,d)$
Decomposes r into $\left(r_{1},r_{0}\right)$ such that $r\;\equiv \;r_{1}2^{d}\;+\;r_{0}\;mod\;q$**Input:** $r\;\in \;Z_{q}$  1: $r^{+}\leftarrow r\;mod\;q$  2: $r_{0}\leftarrow r^{+}\;{mod}^{\pm }\;2^{d}$  3: **return** (($r^{+}-r_{0}$)$∕2^{d}$, $r_{0}$)**Output:** $\left(r_{1},\;r_{0}\right)$

Algorithm 2 is utilized for selecting α as a divisor of q − 1. This alternative methodology then permits the decomposition to be expressed as $r=\;r_{1}\cdot \alpha +r_{0}$.
**Algorithm 2** 
Decompose(r,α)
**Input:** r, α  1: $r:=\;r\;{mod}^{+}\;q$  2: $r_{0}\;:=\;r\;{mod}^{\pm }\;\alpha$  3: **if** $r-r_{0}=q-1$  4:  **then** $r_{1}\;:=0;\;r_{0}\;:=r_{0}-1$  5: **else** $r_{1}\;:=(r\;-r_{0})/\alpha$**Output:** $\left(r_{1},\;r_{0}\right)$

Subsequently, the extraction of these respective higher-order $r_{1}$ and lower-order $r_{0}$ bit components from the element r is accomplished using Algorithms 3 and 4.
**Algorithm 3** 
HighBits (r,α)
**Input:** r, α  1: $\left(r_{1},\;r_{0}\right):=Decompose(r,\alpha )$**Output:** $r_{1}$
**Algorithm 4** 
LowBits (r,α)
**Input:** r, α  1: $\left(r_{1},\;r_{0}\right):=Decompose(r,\alpha )$
**Output:** $r_{0}$


**Lemma 1.** *Assuming ${\left||s|\right|}_{\infty }\leq \;\beta$* *and ${\left||LowBits(r,\;\alpha )|\right|}_{\infty }<\alpha /2-\beta$. Then we consider the following formula to hold: HighBits(r,α) =HighBits(r+s, α)*.


### 5.3. The Proposed Scheme

Our scheme is based on the contributions of Ducas [30]. Parameter settings are as follows: Let the matrix elements be polynomials in the ring $R_{q}=\;Z_{q}\left[X\right]/(X^{n}+1)$. Positive integers d specifically indicate the bit length for element decomposition and are invariably set to d = 13. For the modulus q, we choose $q=2^{23}-2^{13}+1$, while the polynomial degree n is fixed at 256. A critical constraint is that coefficients of all sampled random vectors must not exceed η. We define $\gamma_{1}=\left(q-1\right)∕16$, $\gamma_{2}=\gamma_{1}∕2$, and $b=\lceil log_{2}\eta \rceil$. The parameter β denotes the maximum allowable magnitude for coefficients in vectors considered to have ‘small’ entries. The construction of the proposed scheme is outlined below: Algorithms 5–10 constitute the primary signature and verification, while Algorithms 11–13 constitute the re-signature and verification.
**Algorithm 5 Setup:**1:  $\rho \leftarrow \{0,1{\}}^{256}$2:  $\rho^{\prime }\leftarrow \{0,1{\}}^{512}$3:  $K\leftarrow \{0,1{\}}^{256}$4:  $k\times l\;matrix\;A≔ExpandA\left(\rho \right)$5:  $\left(s_{1},s_{2}\right)\leftarrow S_{\eta }^{l}\times S_{\eta }^{k}≔ExpandS\left(\rho^{\prime }\right)$6:  $t≔As_{1}+s_{2}$7:  $pk=\left(\rho ,t\right),sk=\left(\rho^{\prime },K,s_{1},s_{2}\right)$8:  $H_{1}:\{0,1{\}}^{*}\times Z_{q}^{k}\times Z_{q}^{k}\to \{0,1{\}}^{*}$9:  $H_{2}:\{0,1{\}}^{*}\times \{0,1{\}}^{*}\times Z_{q}^{k}\times Z_{q}^{k}\to \{0,1{\}}^{*}$
**Algorithm 6 Partial Private Key Extract:**10: User submits identity information ID11: KGC selects  $\rho_{1}\leftarrow \{0,1{\}}^{256}$12:  $r_{1}\in S_{\gamma_{1}-1}^{l}≔ExpandMask\left(K∥\rho_{1}\right)\;$13:  $R≔Ar_{1}$14:  $\left(R_{1},R_{0}\right)≔HighBits\left(R,2\gamma_{2}\right)$15:  $\tilde{c_{1}}\in \{0,1{\}}^{256}=H_{1}\left(ID∥t∥R_{1}\right)$16:  $c_{1}\in B_{60}:=SampleInBall\left(\tilde{c_{1}}\right)$17: partial private key  $d_{a}≔c_{1}s_{1}+r_{1}$18:  $r_{0}=Lowbits\left(R-c_{1}s_{2},2\gamma_{2}\right)$19: **If**
$∥d_{a}{∥}_{\infty }\geq \gamma_{1}-\beta \;$ **or** $∥r_{0}{∥}_{\infty }\geq \gamma_{2}-\beta$, then return to step 1020: KGC sends the partial private key $\left(d_{a},c_{1}\right)$ to user21: User calculates $R_{1}^{\prime }:=HighBits\left(Ad_{a}-c_{1}t,2\gamma_{2}\right)$22: **If**
$c_{1}^{\prime }:=H_{1}\left(ID∥t∥R_{1}^{\prime }\right)$ is determined to be invalid23:  **then** user rejects it and initiates to the KGC for the partial private keyOutput: $d_{a},c_{1}$
**Algorithm 7 Set Secret Value:**1:  $\rho_{2}\leftarrow \{0,1{\}}^{256}$2:  $K_{1}\leftarrow \{0,1{\}}^{256}$3:  $x_{a}\leftarrow S_{\eta }^{k}$Output: $x_{a}$
**Algorithm 8 Generate full public-private key pairs for users:**4:  $\left(d_{a1},\;d_{a0})\;∶=\;Power2Roundq{(d}_{a},b\right)$5:  $w_{a}:=Ad_{a0}+x_{a}$6:  private key $sk_{a}=\left(K_{1},\rho_{2},d_{a0},x_{a}\right)$7: public key $pk_{a}=w_{a}$
**Algorithm 9 Signature Generation:**Input:  $M,\;ID,sk_{a}\;$8:  $A≔ExpandA\left(\rho \right)$9:  $\mu ≔CRH\left(\rho ∥M\right)$10:  
$y\in S_{\gamma_{1}-1}^{l}≔ExpandMask\left(K_{1}∥\rho_{2}∥\mu \right)$
11:  Y≔Ay12:  $Y_{1}=HighBits\left(Y,2\gamma_{2}\right)$13:  $\tilde{c_{2}}\in \{0,1{\}}^{256}=H_{2}\left(ID∥\mu ∥t∥Y_{1}\right)$14:  $c_{2}\in B_{60}:=SampleInBall\left(\tilde{c_{2}}\right)$15:  $z_{TU}≔c_{2}d_{a0}+y$16:  $r_{3}=Lowbits\left(Y-c_{2}x_{a},2\gamma_{2}\right)$17:  **If**
$∥z_{TU}{∥}_{\infty }\geq \gamma_{1}-\beta$
**or**
$∥r_{3}{∥}_{\infty }\geq \gamma_{2}-\beta$, then return to step 9Output: $\sigma_{TU}=\left(z_{TU},c_{2}\right)$
**Algorithm 10 Verification:**Input: $\sigma_{TU},\;M,\;ID,{pk}_{a}\;$1:  $A≔ExpandA\left(\rho \right)$2:  $\mu ≔CRH\left(\rho ∥M\right)$3:  $Y_{1}^{\prime }=HighBits\left(Az_{TU}-c_{2}w_{a},2\gamma_{2}\right)$4:  **If**
$∥z_{TU}{∥}_{\infty }<\gamma_{1}-\beta$
**and**
c2′=H2ID∥μ∥t∥Y1′Output 1 **else** Output 0

It should be noted that the core part of the signature algorithm consists of a rejection sampling loop, where each iteration of the cycle produces either a valid signature or an invalid signature; the loop repeats until a valid signature is generated and output. The rejection sampling loop follows the Fiat-Shamir with aborts paradigm [17]. The standardization document [31] specifies that Dilithium2/3/5 (corresponding to ML-DSA-44/65/87) requires average cycle counts of 4.25/5.1/3.85 times, respectively, for valid signature generation, as calculated from Equation 5 in [30]. Before requesting shared TU data from CS, HA must first register with KGC to obtain system parameters, as well as both its public ${pk_{b}=w}_{b}$ and private keys $sk_{b}=\left(K_{1},\rho_{2},d_{b0},x_{b}\right)$. The specific process has been described previously.
**Algorithm 11 Proxy Re-key Generation:**Input: $ID_{TU},\;ID_{HA}\;$1: delegator TU selects a random vector $u\leftarrow S_{\eta }^{k}$2:  $K_{TU\to HA}=d_{a0}+u$3: shares vector u with delegatee HA confidentially4: transmits the $K_{TU\to HA}$ to proxy signer CS via a secure channel5: delegatee HA computes $K_{HA\to TU}=d_{b0}+u$6: transmits the $K_{HA\to TU}$ to proxy signer CS via a secure channelOutput $K_{TU\to HA}$, $K_{HA\to TU}$
**Algorithm 12 Proxy Re-signature:**Input: $\sigma_{TU},\;K_{TU\to HA},K_{HA\to TU}$1: proxy signer CS calculates ${z_{HA}=z}_{TU}+K_{HA\to TU}c_{2}-K_{TU\to HA}c_{2}$Output: $\sigma_{HA}=\left(z_{HA},c_{2}\right)$
**Algorithm 13 Verification:**Input: $\sigma_{HA},\;M,ID,{pk}_{b}\;$1:  $A≔ExpandA\left(\rho \right)$2:  $\mu ≔CRH\left(\rho ∥M\right)$3:  $Y_{1}^{\prime }=HighBits\left(Az_{HA}-c_{2}w_{b},2\gamma_{2}\right)$4:  **If**
$∥z_{HA}{∥}_{\infty }<\gamma_{1}-\beta$
**and**
$c_{2}^{\prime }=H_{2}\left(ID∥\mu ∥t∥Y_{1}^{\prime }\right)$Output 1 **else** Output 0

## 6. Correctness and Security Analysis

### 6.1. Correctness

The proof of the correctness of signature verification is as follows:$$\begin{array}{c} Y_{1}^{\prime }=HighBits_{q}(Az_{i}-c_{2}w_{i},2\gamma_{2})\\ =HighBits_{q}(A(c_{2}d_{i0}+y)-c_{2}w_{i},2\gamma_{2})\\ =HighBits_{q}(Ay+c_{2}({Ad}_{i0}-w_{i}),2\gamma_{2})\\ =HighBits_{q}\left(Y-c_{2}x_{i},2\gamma_{2}\right) \end{array}$$

According to Lemma 1, we know that:$$\begin{array}{c} Y_{1}^{\prime }=HighBits_{q}\left(Y-c_{2}x_{i},2\gamma_{2}\right)\\ =HighBits_{q}\left(Y-c_{2}x_{i}+c_{2}x_{i},2\gamma_{2}\right)\\ =HighBits_{q}\left(Y,2\gamma_{2}\right)\\ =Y_{1} \end{array}$$

The verification procedure for the proxy re-signature’s correctness is detailed as follows: When the proxy CS uses the proxy re-signing key and the signature of the delegator TU on message µ to produce a signature for the delegatee HA, the verification can be computed.$$\begin{array}{c} z_{HA}=z_{TU}+K_{HA\to TU}c_{2}-K_{TU\to HA}c_{2}\\ =c_{2}d_{a0}+y+\left(d_{b0}+u\right)c_{2}-\left(d_{a0}+u\right)c_{2}\\ =c_{2}d_{a0}+y+{c_{2}d}_{b0}+c_{2}u-{c_{2}d}_{a0}-c_{2}u\\ =c_{2}d_{b0}+y \end{array}$$

### 6.2. Security Analysis

As illustrated in Figure 3 and Table 2, Dilithium’s security relies on MSIS/MLWE hardness, which is rooted in the worst-case approximate Shortest Vector Problem (SVP) hardness.

#### 6.2.1. Unforgeability


**Lemma 2.** 

*The signature scheme introduced in this work is resistant to Type I adversaries in the random oracle model, provided that the MSIS problem cannot be solved in polynomial time.*




**Proof.** Suppose a Type I adversary, denoted $A_{1}$, successfully compromises the unforgeability of our signature scheme with a non-negligible advantage within a polynomial time frame. It can then be demonstrated that a challenger C, for the MSIS problem, can be constructed, which is consequently able to solve the MSIS problem with a corresponding non-negligible advantage. Challenger C utilizes the following four lists: $L,L_{H_{2}},L_{s},L_{R}$. L for tracking public—key queries made by $A_{1}$, $L_{H_{2}}$ for tracking hash queries to the random oracle $H_{2}$, $L_{s}$ for tracking signature queries, and $L_{R}$ for tracking proxy re—signature queries. Setup: Challenger C takes the security parameter n as input, computes the system parameters A and the public-private key pair pk=(ρ,t), ${sk=(\rho }^{\prime },K,s_{1},s_{2})$.Queries:(1)Query-of-user’s creation: Challenger C selects an index j randomly, 0≤j≤q, and the aim is to construct a fraudulent signature for the identity $ID_{j}$. Initially, challenger C prepares an empty list L structured as $\left(ID_{i},c_{1i},\rho_{2i},\;K_{1i},\;d_{i0},x_{i},\;w_{i},\;y_{i}\right)$. When $A_{1}$ queries for identity $ID_{i}$, C scans the entire list L to confirm if the public key for the given identity already exists. If it does, C returns the corresponding key $w_{i}$ to $A_{1}$. Otherwise, C randomly chooses $\rho_{2},K_{1},d_{i}\in S_{\gamma 1}^{l}$, and $x_{i}\in S_{\eta }^{k}$, then computes $\left(d_{i1},d_{i0})=Power2Roundq(d_{i},b\right)$, $w_{i}=Ad_{i0}+x_{i}$, and updates the tuple $\left(ID_{i},c_{1i},\rho_{2i},\;K_{1i},\;d_{i0},x_{i},\;w_{i},\;y_{i}\right)$. Finally, C returns $w_{i}$ to $A_{1}$.(2)Query-of-$H_{2}$: Challenger C sets up an empty list $L_{H2}$ = $\left(ID_{i},\;Y_{i1},\mu_{i}\;,c_{2i}\right)$. When $A_{1}$ makes a query for identity $ID_{i}$, C consults the entire list $L_{H2}$; if a corresponding value $c_{2i}$ is found, C returns $c_{2i}$ to $A_{1}$. If not, C randomly selects $c_{2}$ and updates the tuple $\left(ID_{i},\;Y_{i1},\mu_{i},c_{2i}\right)$ in list $L_{H2}$, and then returns $c_{2i}$ to $A_{1}$.(3)Query-of-partial-private-key: When $A_{1}$ makes a query of a partial private key for $ID_{i}$, C scans the entire list L. If the list lacks a relevant entry, C executes the user creation algorithm, updates the tuple $\left(ID_{i},c_{1i},\rho_{2i},\;K_{1i},\;d_{i0},x_{i},\;w_{i},\;y_{i}\right)$ in *L*, and returns $d_{i0}$ to $A_{1}$. If *i = j*, C aborts the query.(4)Query-of-replace-public-key: When $A_{1}$ wants to replace the public key $\left(ID_{i},w_{i}\right)$ of identity $ID_{i}$ with $\left(ID_{i},w_{i}^{\prime }\right)$. C scans the entire list L. If C finds the entry, then replace $w_{i}$ with $w_{i}^{\prime }$. If not, C performs the user’s creation, subsequently updates list L with the generated tuple $\left(ID_{i},c_{1i},\rho_{2i},\;K_{1i},\;d_{i0},x_{i},\;w_{i},\;y_{i}\right)$, then replaces $w_{i}$ with $w_{i}^{\prime }$.(5)Query-of-secret-value: Upon adversary $A_{1}$’s query of secret value for identity $ID_{i}$, challenger C inspects the entire list L. If a matching entry is found, C provides the associated tuple $\left(\rho_{2i},\;K_{1i},\;d_{i0},x_{i}\right)$ to $A_{1}$. If no such entry exists, C executes the user’s creation, adds the new tuple $\left(ID_{i},c_{1i},\rho_{2i},\;K_{1i},\;d_{i0},x_{i},\;w_{i},\;y_{i}\right)$ to L, and subsequently returns the resultant $\left(\rho_{2i},\;K_{1i},\;d_{i0},x_{i}\right)$ to $A_{1}$.(6)Query-of-proxy-pe-key: C initializes an empty list $L_{R}$ as $\left(ID_{i},\;ID_{j},rk_{i\to j},rk_{j\to i}\right)$. When $A_{1}$ sends $(ID_{i},ID_{j}$) to C, C checks $L_{R}$ and sends the corresponding proxy re-key to $A_{1}$.(7)Query-of-signature: When $A_{1}$ queries the signature for identity $ID_{i}$ and message Mi, C checks list $L_{s}$. If the entry exists, C returns $\left(z_{i},\;Y_{i1},\;c_{2i}\right)$ to $A_{1}$. Otherwise, C runs the user creation algorithm, computes $\left(z_{i},\;Y_{i1}\right)$, updates the tuple, and returns $\left(z_{i},\;Y_{i1},\;c_{2i}\right)$ to $A_{1}$.(8)Query-of-re-signature: $A_{1}$ sends $\left(ID_{i},\;ID_{j},\;M_{j},\;sig_{j}=(z_{j},\;c_{2j})\right)$ to C, asking C to derive a signature for $M_{j}$ under $ID_{j}$ using the signature $sig_{j}=(z_{j},\;c_{2j})$ of $ID_{i}$. C first verifies $sig_{j}=(z_{j},\;c_{2j})$. If valid, C performs a signature query for $\left(ID_{i},\;M_{j}\right)$ and returns the result to $A_{1}$; otherwise, the query is aborted.Forgery: Upon termination of the query phase, the adversary $A_{1}$ outputs a candidate forgery $\left(ID^{*},\;\;M^{*},\;\;sig\right)$. The challenger C first checks if $ID^{*}\neq ID_{j}$, C aborts. If $ID^{*}=ID_{j}$, C deems the forgery successful if the following conditions are simultaneously met:Verify$\left(M^{*},sig\right)=1$;$ID^{*}$ was never submitted by $A_{1}$ to the Query-of-partial-private-key;$ID^{*}$ and $M^{*}$ were never queried in the Quire-of-signature.If adversary $A_{1}$ achieves a valid forgery sig under these conditions, the subsequent equation is then reformulated as:$$HighBits\left(Az-c_{2}w,\;2\gamma_{2}\right)=Az-c_{2}w+u$$
where ||u||∞ ≤ 2γ_2_ + 1. Thus, an adversary $A_{1}$, including quantum adversaries, capable of forging new messages, can find $z,c_{2},u,M$ that satisfy the equation:$$H_{2}\left(\left[A\left|w\right|I_{k}\right]\left[\begin{array}{c} z\\ -c_{2}\\ u \end{array}\right]∥M∥ID∥t\right)=c_{2}$$Invoking the Forking Lemma [32], adversary $A_{1}$ can obtain a novel set of valid signatures $\left(z^{*},\;u^{\prime },\;M\right)$. Thus, the following equations are obtained:$$H_{2}\left(\left[A\left|w\right|I_{k}\right]\left[\begin{array}{c} z\\ {-c}_{2}\\ u \end{array}\right]∥M∥ID∥t\right){=H}_{2}\left(\left[A\left|w\right|I_{k}\right]\left[\begin{array}{c} z^{*}\\ -c_{2}\\ u^{\prime } \end{array}\right]∥M∥ID∥t\right)$$
We can rewrite the following equation as follows:$$\left[A\left|w\right|I_{k}\right]\left[\begin{array}{c} z-z^{*}\\ c_{2}-c_{2}\\ u-u^{\prime } \end{array}\right]=0$$Let$$\left[\begin{array}{c} z-z^{*}\\ c_{2}-c_{2}\\ u-u^{\prime } \end{array}\right]=v$$Challenger C, through interaction with adversary $A_{1}$, thus effectively solves the MSIS hard problem, yielding v as its solution. □



**Lemma 3.** 
*Assuming that the MSIS problem remains computationally difficult within polynomial time in the Random Oracle Model, our proposed signature scheme achieves existential unforgeability against Type II adversaries.*




**Proof.** Suppose a Type II adversary $A_{2}$, breaks the existential unforgeability of our scheme in polynomial time with non-negligible advantage. A challenger C, tasked with simulating an MSIS problem instance, can then be constructed to solve the MSIS problem with a comparable non-negligible advantage by utilizing $A_{2}$. Challenger C utilizes the following four lists: $L,L_{H_{2}},\;L_{s},L_{R}$. L for tracking public—key queries made by $A_{1}$, $L_{H_{2}}$ for tracking hash queries to the random oracle $H_{2}$, $L_{s}$ for tracking signature queries, and $L_{R}$ for tracking proxy re-signature queries. Lemma 3’s proof methodology resembles Lemma 2’s, differing mainly in the adversarial capabilities of the adversary $A_{2}$. Specifically, a Type II adversary cannot issue Secret Value Queries nor Public Key Replacement Queries for the target identity whose signature is to be forged. Due to space constraints, a detailed elaboration is omitted here. Through a sequence of oracle interactions and application of the Forking Lemma [32], $A_{2}$ eventually acquires $\left(z,\;c_{2},u,M\right)$. These values fulfill the following equation:$$H_{2}\left(\left[A\left|w\right|I_{k}\right]\left[\begin{array}{c} z\\ {-c}_{2}\\ u \end{array}\right]∥M∥ID∥t\right)=c_{2}$$By virtue of the Forking Lemma [32], it follows that adversary $A_{2}$ can obtain a novel, valid signature tuple $\left(z^{\prime },c_{2}^{\prime },u^{\prime },M\right)$. Consequently, the following equations are satisfied:$$H_{2}\left(\left[A\left|w\right|I_{k}\right]\left[\begin{array}{c} z\\ {-c}_{2}\\ u \end{array}\right]∥M∥ID∥t\right){=H}_{2}\left(\left[A\left|w\right|I_{k}\right]\left[\begin{array}{c} z^{\prime }\\ {-c}_{2}^{\prime }\\ u^{\prime } \end{array}\right]∥M∥ID∥t\right)$$
We can rewrite the following equation as follows:$$\left[A\left|w\right|I_{k}\right]\left[\begin{array}{c} z-z^{\prime }\\ c_{2}^{\prime }-c_{2}\\ u-u^{\prime } \end{array}\right]=0$$Let$$\left[\begin{array}{c} z-z^{\prime }\\ c_{2}^{\prime }-c_{2}\\ u-u^{\prime } \end{array}\right]=v$$Challenger C, through interaction with adversary $A_{2}$, thus effectively solves the MSIS hard problem, yielding v as its solution. □


#### 6.2.2. Security Against Key Recovery Attacks

The setup process of the Key Generation Center (KGC) and the private key generation process of the user within our proposed scheme are analogous to the GEN algorithm employed in Dilithium [30]; their security is predicated on the Module Learning with Errors (MLWE) assumption. This assumption implies that a public key, formed as $\left(A,t={As}_{1}+s_{2}\right)$, is computationally indistinguishable from a pair $\left(A,t^{\prime }\right)$ where $t^{\prime }$ is a uniformly random element. Consequently, our scheme provides robust protection against key recovery attacks under the assumption of MLWE hardness.


**Definition 4.** 

*Existential Unforgeability under Chosen Message Attack (EUF-CMA) [33]: A signature scheme achieves EUF-CMA security in the Random Oracle Model if, for any polynomial-time adversary A, the probability that A wins the standard EUF-CMA security game is negligible. This security notion ensures that an adversary cannot forge a valid signature for any message for which they have not previously obtained a signature by querying the legitimate signing oracle.*




**Definition 5.** 

*Lemmas 2 and 3 collectively establish the resistance of our scheme to Type I and Type II adversaries, thereby proving its EUF-CMA security.*



## 7. Performance Evaluation

### 7.1. Computation Cost

In this section, we provide a comparative analysis of the computational costs associated with our proposed scheme and those of existing lattice-based certificateless proxy re-signature schemes [12,27]. The evaluation framework is adapted from the methodology established by Yu et al. [34] and incorporates enhancements from the computational techniques proposed by Xu et al. [35]. Performance benchmarking was performed on a Windows 11 system featuring an Intel(R) Core(TM) i7-8750H CPU operating at 2.20 GHz. All empirical data were acquired via the PQClean and PALISADE libraries, and the execution time for each cryptographic operation was reported as the arithmetic mean over 1000 iterations. We focused our analysis on the most computationally intensive operations: hashing, Gaussian sampling, preimage sampling, matrix-vector multiplication, and matrix-vector addition. Given that Dilithium2 satisfies NIST Security Level 2, offering collision resistance on par with SHA-256 and classical security surpassing AES-128, we configured our scheme with the initial parameters $q=2^{23}-2^{13}+1$ and n=256. Table 3 and Table 4 lists the timing results for these fundamental operations within the schemes under comparison.

To provide a clear and quantitative comparison, Table 3 and Table 5 contrast our proposed scheme with prior certificateless lattice-based proxy re-signature schemes, including those of Fan et al. [12] and Zhou et al. [27], across multiple dimensions from algorithm design to performance metrics. For scheme [12], key generation requires one hash operation ($T_{h}$), three matrix-vector multiplication ($M_{v}$), one matrix-vector addition ($M_{a}$) and one original image sampling algorithm ($S_{M}$). Signing requires one Gaussian sampling operation ($S_{G}$), one hash operation, two matrix-vector multiplication, and one matrix-vector addition. Its verification phase entails two hash operations, two matrix-vector multiplications, and one matrix-vector addition. Re-signing requires four matrix-vector additions. Consequently, the key generation cost for scheme [12] is $S_{M}+T_{h}+{3M}_{v}+M_{a}\approx 274.37\;\mu s$, the signing and verification costs are approximately $S_{G}+T_{h}+{2M}_{v}+M_{a}\approx 44.06\;\mu s$, and $2T_{h}+{2M}_{v}+M_{a}\approx 42.05\mu s$, the re-signing cost is ${4M}_{a}\approx 3.56\;\mu s$.

The operation required in the key generation phase of scheme [27] is the same as that of scheme [12]. Scheme [27] necessitates one Gaussian sampling operation, one hash operation, and four matrix-vector multiplications for signing, while its verification phase involves two hash operations, three matrix-vector multiplications, and one matrix-vector addition. In addition, its re-signature phase requires five matrix-vector multiplications and two matrix-vector additions. Thus, the key generation cost for scheme [27] is $S_{M}+T_{h}+{3M}_{v}+M_{a}\approx 274.37\;\mu s$, the signing cost is $S_{G}+T_{h}+{4M}_{v}\approx 80.65\;\mu s$, the verification cost is ${2T}_{h}+3M_{v}+M_{a}\approx 60.79\;\mu s$, and the re-signature cost $5M_{v}+{2M}_{a}\approx 95.48\;\mu s$.

Our proposed scheme requires two hash operations, four matrix-vector multiplications, and four matrix-vector additions for key generation. The signing phase utilizes two hash operations, one matrix-vector multiplication, and two matrix-vector addition operations. The verification process involves one hash operation, one matrix-vector multiplication, and one matrix-vector addition operation, and the re-signature phase requires four matrix-vector addition operations. Therefore, our scheme’s key generation cost is $2T_{h}+{4M}_{v}+4M_{a}\approx 82.2\;\mu s$. Signing cost is $2T_{h}+M_{v}+2M_{a}\approx 24.2\;\mu s$, with the verification and re-signature costs calculated as $T_{h}+M_{v}+M_{a}\approx 21.47\mu s$ and ${4M}_{a}\approx 3.56\;\mu s$.

A comparative summary of computational expenditure for our scheme and others is presented in Table 5 and Figure 4. As shown in Table 4, the Gaussian and preimage sampling algorithms exhibit low efficiency. Furthermore, the TrapGen algorithm can complicate scheme deployment and is computationally intensive. Existing schemes [12,27] employ the TrapGen, preimage sampling, and Gaussian sampling algorithms, thereby consuming substantial resources at the KGC. By eliminating the requirement for these specific algorithms, our proposal facilitates a more straightforward KGC deployment of lattice-based cryptographic systems and enhances the efficiency of its setup procedure. Consequently, as the foregoing analysis indicates, our scheme offers significant advantages in terms of computational overhead.

### 7.2. Data Storage Efficiency

In this section, we detail the storage scheme. Table 6 lists the data storage sizes.(1)KGC public key pk =(ρ,t) is stored in a bit-packed representation. Specifically, the vector t consists of k polynomials $t_{0},\;\ldots ,t_{k-1}$ with coefficients in $\left\{0,\;\ldots \;,\;2^{23}-1\right\}$. This results in each polynomial occupying 256 ⋅ 23/8 = 736 bytes. Consequently, the KGC public key requires 256/8 + 736k = 32 + 736k bytes.(2)KGC private key $sk=(\rho ,K,s_{1},s_{2})$ is stored in a bit-packed representation. The polynomials in $s_{1}$ and $s_{2}$ have coefficients with maximum infinity norm η. Thus, each coefficient is in $\left\{-\eta ,\;\ldots \eta \right\}$. In this scheme, we select η from $\left\{2^{2}+1,\;\ldots \;,\;2^{4}-1\right\}$. This results in each polynomial occupying 256·4/8=128 bytes. Consequently, the KGC private key requires 64+128·k+128·l=64+128k+128l bytes.(3)User’s full public key $pk_{a}=w$ is stored in a bit-packed format. Specifically, the vector w consists of k polynomials $w_{0},\ldots ,w_{k-1}$ with coefficients in $\left\{0,\ldots ,2^{23}-1\right\}$. This results in each polynomial occupying 256·23/8=736 bytes. Consequently, the user’s full public key requires 736k bytes.(4)User’s full private key contains $sk_{a}=(K_{1},\rho_{2},d_{a0},x_{a})$ and is stored in a bit-packed format. The polynomials in $d_{a0}$ have coefficients in $\left\{2^{2}+1,\ldots ,2^{4}-1\right\}$, resulting in each polynomial occupying 256·4/8=128 bytes. The polynomials in $x_{a}$ have coefficients bounded in infinity norm by η, meaning each coefficient is in {−η, …η}, also resulting in each polynomial occupying 256·4/8=128 bytes. Thus, the user’s full private key requires 64+128l+128k bytes.(5)User’s authorization information u is stored in a bit-packed format. The polynomials in u have coefficients bounded in infinity norm by η, meaning each coefficient is in {−η, …η}, also resulting in each polynomial occupying 256·4/8=128 bytes.(6)The signature $\sigma =\left(z,c_{2}\right)$ is stored in a bit-packed format. The coefficients of the z polynomial are in $\left\{-\gamma_{1}+1,\ldots ,\gamma_{1}-1\right\}$, resulting in each polynomial being 256·20/8 = 640 bytes. $c_{2}$ requires 32 bytes. Thus, the signature requires 640l + 32 bytes.

Table 7 presents the parameters for the specific instances. Following the parameters of Dilithium, we adjusted the parameters to better suit our scheme. In terms of actual performance, the three instances demonstrate an average rejection rate of 1.3–1.4 iterations per signature, which is within the acceptable range of 1.0–3.0. Furthermore, all signatures are generated within five iterations, safely under the upper limit that requires 99.9% of signing attempts to complete in under 10 iterations. It is also evident that even for the third instance, which models the worst-case scenario for NIST Security Level 2, the combined key and signature sizes remain below 5 KB.

Table 8 presents a comparison between our proposed scheme and what is considered among the most efficient existing schemes based on NTRU lattices [36]. We posit that signature length is a critical factor influencing both storage and communication performance. The structural properties of NTRU lattices generally offer advantages over algebraic lattices in terms of storage size. Nevertheless, when compared at the same security level, our scheme achieves smaller signature sizes, with a reduction of 52.9%. Furthermore, owing to the inherent advantages of algebraic lattices, our scheme provides enhanced security and greater ease of implementation.

Furthermore, because the scheme [36] is based on NTRU lattices, it suffers from several drawbacks [30]. For instance, it necessitates sampling from discrete Gaussian distributions, which typically precludes constant-time implementations. The second disadvantage is that its security relies on the NTRU assumption rather than on problems like Ring Learning With Errors (RLWE) or Module Learning With Errors (MLWE). This is crucial because Kirchner and Fouque [37] leveraged the NTRU lattice structure to execute a markedly stronger attack on the big-mode/small-secret version of the problem. Consequently, our approach demonstrates superior performance in terms of both security and computational overhead.

## 8. Limitations and Future Work

While the proposed scheme exhibits significant advantages in terms of post-quantum security, certificate-less nature, and computational and storage efficiency over comparable lattice-based schemes, its widespread deployment in resource-constrained environments, such as the Internet of Things (IoT), necessitates acknowledging its inherent limitations.

First, concerning energy consumption, our performance evaluation indicates that the proposed scheme outperforms existing solutions in terms of computational overhead. Nevertheless, the underlying lattice-based cryptographic operations are inherently computationally intensive. Furthermore, the rejection-sampling mechanism integral to the scheme has a non-constant execution time. Although the average number of repetitions is acceptable, the iteration count for signature generation still exhibits minor variability. This non-determinism can cause fluctuations in the energy consumption of a device, posing a considerable challenge for battery-dependent IoT terminals that demand extended operational lifetimes. Second, the scalability of the system presents another key concern. The system model of the scheme depends on a centralized Key Generation Center (KGC) for the issuance of partial private keys. In large-scale IoT ecosystems comprising millions of devices, this centralized KGC is likely to become a performance bottleneck and a single point of failure. Finally, and of critical importance, this paper details the scheme’s design at the algorithmic level. Although the officially specified Dilithium algorithm is purported to resist side-channel attacks, its resilience on physical hardware is not guaranteed. The potential for key information leakage through side-channel attacks, such as power analysis or timing attacks, requires empirical validation once the scheme is implemented in physical devices. Furthermore, this scheme primarily focuses on the authenticity of identities and the unforgeability of messages in its design, without optimizing privacy features such as unlinkability of signatures or signer anonymity. This may increase privacy leakage risks in highly sensitive application scenarios like electronic health record (EHR) sharing.

In conclusion, our future work will focus on three core directions: (1) Subsequent work will empirically validate the scheme’s deployment feasibility and efficiency advantages on resource-stringent end devices through concrete IoT use cases (e.g., smart health monitoring or environmental sensor networks). (2) Fine-grained hardware-software co-optimization: Researching and developing constant-time implementations of our scheme to eliminate timing channels and flatten energy consumption. (3) Enhanced scalability: Exploring distributed or hierarchical KGC architectures to alleviate centralization bottlenecks. (4) Strengthened security: Formally integrating mature side-channel countermeasures and conditional privacy preservation into the scheme implementation with formal proofs of security.

## 9. Conclusions

This paper introduces an algebraic lattice-based certificateless proxy re-signature scheme based on the design principles of the Dilithium signature scheme [30]. Our construction relies on the cryptographic hardness of the Module Small Integer Solution (MSIS) and Module Learning With Errors (MLWE) problems over algebraic lattices. This foundation facilitates significant reductions in both key and signature sizes, thereby enhancing storage efficiency. The security of the proposed scheme is formally proven using the Random Oracle Model. In conclusion, our scheme offers a relevant pathway for designing certificateless proxy re-signatures aligned with emerging post-quantum standards, potentially enabling its deployment in resource-constrained environments such as IoT devices and Medical Internet of Things (IoMT) data-sharing scenarios.

## Figures and Tables

**Figure 1 sensors-25-04848-f001:**
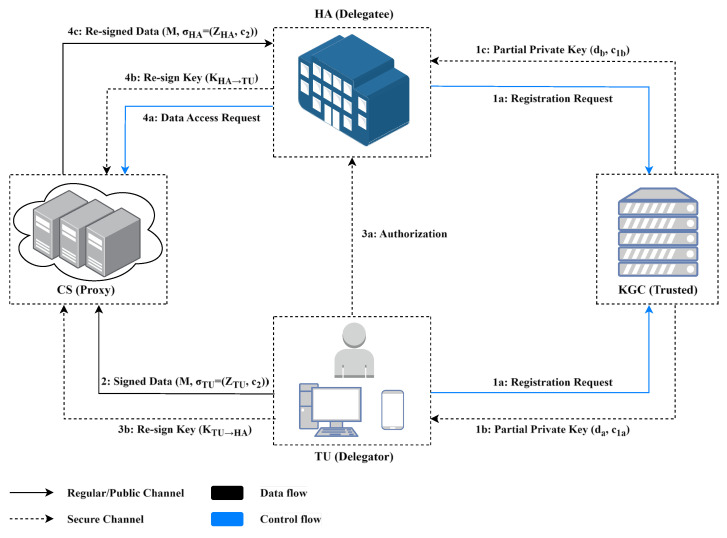
System model of the proposed certificateless proxy re-signature scheme.

**Figure 2 sensors-25-04848-f002:**
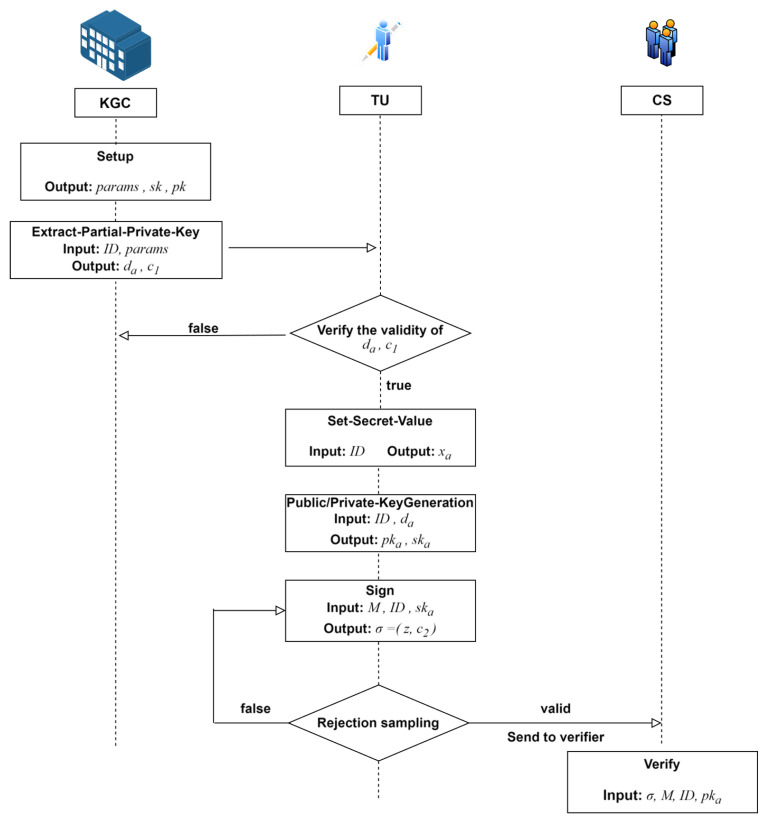
Process flow of TU performing the original signature and CS verifying the signature.

**Figure 3 sensors-25-04848-f003:**
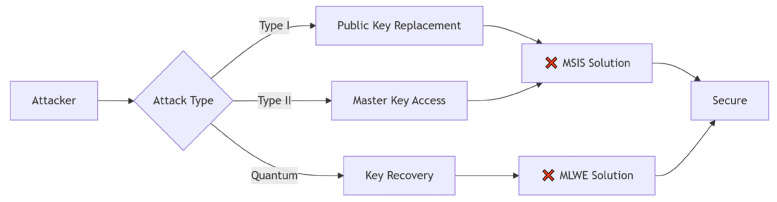
Mapping attack paths to lattice problems.

**Figure 4 sensors-25-04848-f004:**
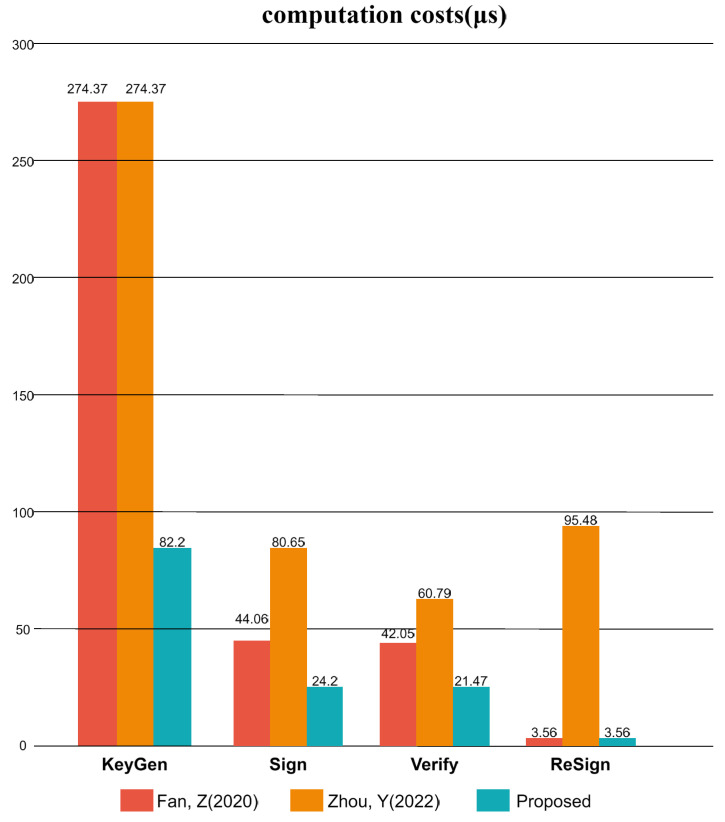
Computation cost [12,27].

**Table 1 sensors-25-04848-t001:** Symbol description.

Notation	Description
q	Prime number
K	A private random seed
k,l	Integers, dimensions of A
η	Private key range
$S_{\eta }^{l}$	the subset of polynomial vectors
τ	The number of ±1 in polynomial c
β	β=τ · η
$\gamma_{1}$	Coefficient range of y
$\gamma_{2}$	Low-order rounding range
Z	The ring of integers
R	Univariate polynomial ring on Z
$B_{\tau }$	The set of all polynomials in $\;R_{q}$
brv	Bit reversal
Bold letter, such as $R_{A}$	Vector
M	Message
${ID}_{i}$	Entities’ identity
pk,sk	Public-private key of KGC
${pk}_{i},{sk}_{i}$	Public and private key of Entities’ identity
σ	Signature

**Table 2 sensors-25-04848-t002:** Overview of the security attributes.

Adversary Type	Capabilities	Assumption	Attack Surface Mitigated
Type I	Replace public keys	MSIS Hardness	Malicious KGC + Key Replacement
Type II	Access master key	MSIS Hardness	Honest-but-Curious KGC
Key Recovery	Eavesdrop on communications	MLWE Hardness	Quantum Key Search Attacks

**Table 3 sensors-25-04848-t003:** Comparison with existing certificateless lattice-based schemes.

Feature	Fan [12]	Zhou [27]	Our Scheme
Special Features	TrapGen-based	Collusion-resistant	Dilithium-based
Core Primitives	TrapGen, Gaussian Sampling, Preimage Sampling	TrapGen, Gaussian Sampling, Preimage Sampling	Rejection Sampling
Hardness Assumptions	SIS, ISIS	SIS, ISIS	MLWE, MSIS
Security Proof	EUF-CMA in Random Oracle Model	Collusion Attack Resistance	EUF-CMA in Random Oracle Model

**Table 4 sensors-25-04848-t004:** Notation and function execution times.

Notation	Description	Function Execution Time
$T_{h}\;$	One-way hash	≈1.84 μs
$S_{M}$	Preimage sampling	≈215.42 μs
$S_{G}$	Gaussian sampling	≈3.85 μs
$M_{v}$	matrix-vector multiplication	≈18.74 μs
$M_{a}$	matrix-vector addition	≈0.89 μs

**Table 5 sensors-25-04848-t005:** Computational expenditure comparison.

Operation	Fan [12]	Zhou [27]	Our Scheme
KeyGen	274.37 μs	274.37 μs	82.2 μs
Sign	44.06 μs	80.65 μs	24.2 μs
Verify	42.05 μs	60.79 μs	21.47 μs
Re-Sign	3.56 μs	95.48 μs	3.56 μs

**Table 6 sensors-25-04848-t006:** Notation and storage size.

Notation	Description	Storage Size (Bytes)
pk	$KGC\;public\;key\;\left(\rho ,t\right)$	32+736k
sk	$KGC\;private\;key\;\left(\rho ,K,s_{1},s_{2}\right)$	64+128k+128l
${pk}_{a}$	$User^{\prime }s\;full\;public\;w$	736k
${sk}_{a}$	$User^{\prime }s\;full\;private\;key\;{(K}_{1},\rho_{2},d_{a0},x_{a})$	64+128l+128k
u	$User^{\prime }s\;authorization\;information\;u$	128k
σ	$Signature\;(z,c_{2})$	640l+32

**Table 7 sensors-25-04848-t007:** Parameters of specific instances.

	Instance 1	Instance 2	Instance 3
(k, l)	(3,2)	(4,4)	(5,6)
η	2	2	4
τ	60	60	60
β	120	120	240
KGC pk (bytes)	2240	2976	3712
KGC sk (bytes)	704	1088	1472
User pk (bytes)	2208	2944	3680
User sk (bytes)	704	1088	1472
u (bytes)	384	512	640
Sig (bytes)	1312	2592	3872

**Table 8 sensors-25-04848-t008:** Comparison of required storage sizes.

	Xu [36]	Our Scheme
Lattice	NTRU Lattice	Algebraic Lattice
Required storage sizes of Sig	2787 bytes	1312 bytes

## Data Availability

Data are contained within the article.

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
