# Peer review of "Lattice-Based Certificateless Proxy Re-Signature for IoT: A Computation-and-Storage Optimized Post-Quantum Scheme"

_sensors, 2025, doi:10.3390/s25154848_

Round 1
Reviewer 1 Report
Comments and Suggestions for Authors
The manuscript presents a promising certificateless proxy re-signature scheme optimized for post-quantum IoT systems. The core idea is valuable and the theoretical construction is sound. However, the paper requires several refinements before it can be considered for publication. Please consider the following specific suggestions:
1. Novel Contribution Using Dilithium – Well-Motivated but Needs Comparative Clarification: “We propose an efficient certificateless proxy re-signature scheme... leveraging the Dilithium algorithm.” Page 3; While the design is novel, its distinction from prior lattice-based schemes such as [12] and [27] is not explicitly quantified.
My recommendation: Provide a clear comparison table (preferably in Section 2 or 6) contrasting your work with existing certificateless lattice-based proxy re-signature schemes in terms of algorithm, security model, key size, and storage/computation tradeoffs.
2. Figure 1 – Lacks Clarity and Technical Labels: “Figure 1. This is a figure. Schemes follow the same formatting.” Page 7; Figure 1 in the manuscript lacks essential scientific structure and clarity, which diminishes the reader's ability to fully understand the proposed system. The figure does not include phase numbering to indicate protocol steps (e.g., registration, key generation, re-signature), and the arrow labels are vague—terms like "Authorization" or "Send Signature" fail to specify whether the message contains keys, signatures, or control commands. Moreover, the figure does not distinguish between data flow (e.g., digital signatures) and control flow (e.g., registration requests), nor does it indicate whether communication channels are secure or trusted. The roles of entities in generating or storing the re-signature key are also visually unclear.
My recommendation: To improve the diagram, it is recommended to redraw it with numbered arrows indicating protocol phases (e.g., Phase 1, Phase 2). The message contents should be explicitly labeled (e.g., signature, re-signature key, partial key), and arrows can be color-coded to distinguish message types (registration, data, keys). It is also advisable to indicate secure vs. insecure channels and include a clear title and legend to explain all components (KGC, TU, CS, HA). These enhancements are essential for presenting a scientifically accurate and readable model.
3. No Concrete Use Case Demonstrating IoT Application: “...offering marked advantages for deployment in storage-constrained environments, including IoT terminals.” Page 3; The claim of suitability for IoT is not supported by a use case, real-world dataset, or prototype deployment.
My recommendation: Include a simulated or real-world IoT application (e.g., smart health, environmental sensors) demonstrating feasibility and efficiency under resource constraints.
4. No Performance Tables or Visual Benchmarks: “Performance benchmarking was executed on a Windows 11 system...” Page 15; No tables or figures were presented to substantiate performance claims (e.g., compression rate, operation times).
My recommendation: Include a performance evaluation table with: (Time for KeyGen, Sign, Verify, ReSign (ms)), Signature/key sizes (bytes), and Comparison with [12], [27].
5. Storage Compression Not Quantified Theoretically: “...a storage space compression rate of over 52.9%...” Page 3; There’s no derivation or formula showing how the 52.9% compression was achieved.
My recommendation: Provide a formal derivation and define exactly what is being compressed (signature only? keys as well?). Include size-before and size-after comparisons.
6. Security Analysis Too Textual - Needs Summary Tables: “.... our proposed signature scheme achieves existential unforgeability against", Type I and "Type II adversaries.” Page 14; The security proofs are lengthy and dense, lacking visual aids.
My recommendation: Add a summary table showing: Adversary Type, Assumption, Resistance Model, and Attack Surface Mitigated.
7. No Mention of Key Size Details: There is no specific mention of key size (in bits or bytes), which is crucial when optimizing for IoT.
My recommendation: Include a table specifying; Public/private key sizes (in bytes), Signature size, and How this compares to NTRU, Dilithium, and other lattice schemes.
8. Lack of Operational Complexity Analysis: “...achieving a 52.9% reduction in storage space.” Page 1; There’s no analysis of time/space complexity (e.g., O(n), O(n log n)).
My recommendation: Provide asymptotic analysis of major functions, especially signing and verification, and show whether the improvements are linear, sublinear, etc.
9. Rejection Sampling Not Analyzed for Practical Convergence: “...each iteration of the cycle produces either a valid signature or an invalid signature...” Page 11; There is no mention of expected number of iterations or practical failure rates.
My recommendation: Include the probability of success per iteration, and average number of attempts until success, possibly in a Monte Carlo simulation.
10. No Explicit Discussion of Limitations; The manuscript lacks a section addressing known or potential limitations.
My recommendation: Include a paragraph in the conclusion or a separate “Limitations and Future Work” section discussing: Energy consumption, Scalability to large-scale systems, and Absence of side-channel resistance.
11. Dilithium Assumptions Not Critically Evaluated: “...based on the dual hardness assumptions of the MSIS and MLWE problems.” Page 3; No discussion is provided on the current state of MSIS/MLWE assumptions in literature (e.g., known weaknesses or parameter tightening).
My recommendation: Discuss recent findings (2022–2024) regarding these assumptions, and whether any known attacks affect specific parameter sets.
12. Algorithms Lack Pseudocode Formatting: “Algorithm 1 Power2Roundq (r, d) …” Page 9; The algorithms are written in a dense paragraph format, lacking clarity.
My recommendation: Format algorithms in standard pseudocode with: Clear input/output, Step numbers, and Proper indentation and spacing.
13. No Open Source Code or Implementation for Validation; No software implementation is mentioned, making it difficult for others to verify or extend the work.
My recommendation: Provide a link to a prototype (e.g., GitHub) or mention plans to open-source the code under a permissive license.
14. Privacy Considerations Not Addressed; While the paper emphasizes authenticity and unforgeability, it does not discuss privacy (e.g., traceability of signatures or unlinkability).
My recommendation: Briefly discuss whether the scheme supports privacy-preserving re-signatures or if future work will address that (e.g., through chameleon hashes or ring signatures).
Reviewer 2 Report
Comments and Suggestions for Authors
This paper constructs an efficient post-quantum certificateless proxy re-signature scheme based on algebraic lattices. It is of great research value.
Hower, It is necessary to provide a security analysis and assessment.
Reviewer 3 Report
Comments and Suggestions for Authors
Overall, the paper presents an algebraic lattice-based certificateless proxy re‑signature scheme grounded in MSIS/MLWE hardness, offering formal EUF‑CMA proofs and efficient performance. Minor clarifications on notation consistency, rejection sampling parameters, acronym definitions, and caption contexts will enhance clarity. Overall, a valuable contribution to post‑quantum cryptography design, with practical deployment considerations included. the paper is suitable for publication after these revisions:
improve the rational used in in MSIS/MLWE assumptions with formal security proofs.
Comprehensive performance evaluation demonstrating reduced signing and verification costs.
Clarify notation consistency—e.g. distinguish γ₁/γ₂ versus β across sections.
Improve description of the rejection‑sampling loop parameters and expected iteration counts.
Add explicit definitions for all acronyms (e.g. TU, CS) upon first use.
Enhance figure and table captions with brief methodological context.
Briefly discuss practical deployment scenarios, including side‑channel considerations.
Round 2
Reviewer 1 Report
Comments and Suggestions for Authors
I sincerely appreciate your thorough and well-structured responses to all reviewer comments. The manuscript has been significantly improved in terms of clarity, technical rigor, and structure. I commend your efforts, especially in incorporating comparative tables, enhancing the figures, and clarifying the security model. I recommend that, before final acceptance, you perform a minor English language polishing, particularly in some narrative sections such as the Future Work and Outlook paragraph, to further improve fluency and readability.